# Expression of Aquaglyceroporins in Spermatozoa from Wild Ruminants Is Influenced by Photoperiod and Thyroxine Concentrations

**DOI:** 10.3390/ijms23062903

**Published:** 2022-03-08

**Authors:** Julián Santiago-Moreno, Belén Pequeño, Belen Martinez-Madrid, Cristina Castaño, Paula Bóveda, Rosario Velázquez, Adolfo Toledano-Díaz, Manuel Álvarez-Rodríguez, Heriberto Rodríguez-Martínez

**Affiliations:** 1Department of Animal Reproduction, Spanish National Institute for Agricultural and Food Research and Technology (INIA-CSIC), 28040 Madrid, Spain; pequeno.belen@inia.es (B.P.); cristina.castano@inia.es (C.C.); paubgbio@gmail.com (P.B.); rosario.velazquez@inia.es (R.V.); toledano@inia.es (A.T.-D.); 2Department of Animal Medicine & Surgery, Faculty of Veterinary Medicine, Universidad Complutense de Madrid (UCM), 28040 Madrid, Spain; belen.martinez@vet.ucm.es; 3Department of Biomedical & Clinical Sciences (BKV), Obstetrics & Gynecology, Linköping University, SE-58185 Linkoping, Sweden; manuel.alvarez-rodriguez@liu.se (M.Á.-R.); heriberto.rodriguez-martinez@liu.se (H.R.-M.)

**Keywords:** Iberian ibex, mouflon, aquaglyceroporins, melatonin, sperm cryoresistance, thyroxine

## Abstract

This work identified the presence of AQPs in frozen-thawed sperm of wild ruminants and assessed the influence of the interaction between photoperiod and thyroxine on AQP expression, and on testosterone secretion. Thyroxine and melatonin were administered to ibexes. In a second experiment, performed in mouflons, circulating thyroxine was reduced via treatment with propylthiouracil (PTU), and an artificial long day (LD) photoperiod established. In the ibexes, the melatonin treatment increased the blood plasma testosterone concentration, reduced the cryoresistance ratio (CR) for sperm viability and the presence of an intact acrosome, and increased the percentage of sperm with AQP7 in the acrosome and of AQP3 and AQP10 in the midpiece. In the mouflons, neither the PTU treatment, the LD, nor the combination of both affected the CR of any sperm variable. The percentage of sperm with AQP3 increased in the post-acrosome region but decreased in the tail in the LD+PTU group. The percentage of sperm with AQP10 in the principal piece and endpiece was lower in the PTU+LD group than in the control and LD groups. The influence of photoperiod/melatonin on AQP expression might be indirectly exerted through changes in the testosterone concentration, and thus ultimately affect sperm cryoresistance.

## 1. Introduction

The annual variation in ruminant sperm cryoresistance is related to seasonal changes in testosterone and prolactin [1,2,3]. Indeed, wild ruminant sperm shows its greatest cryoresistance at the end of the rutting season when testosterone levels fall. Other hormones involved in the modulation of the breeding season might, therefore, also be involved in the annual variation in sperm freezability. The major environmental cue regulating seasonal breeding activity in ruminants is the photoperiod [4], with testicular activity and spermatogenesis reaching their maximum during the autumn when day length is shortening [5,6]. The photoperiodic signal is transduced by the pineal gland into a pattern of melatonin secretion which, in turn, provides a critical endocrine signal for regulating the secretion of other hormones involved in the onset and termination of the breeding season [7]. Certainly, melatonin is present in ram seminal plasma [8], and the melatonin receptors MT1 and MT2 have been identified in the sperm plasma membrane [9]. The possible role of melatonin in improving sperm cryoresistance is, however, controversial. Neither treatment with melatonin implants nor the establishment of an artificial short day-photoperiod improves the quality of cooled or frozen-thawed buck sperm [10], yet the supplementation of semen extender with melatonin protects ram spermatozoa from cryopreservation injuries in a dose-dependent manner [11]. 

Certainly, the thyroid hormones play a key role in the control of seasonal reproduction in ruminants [12,13]. Indeed, the presence of thyroid hormones is decisive in the expression of photo-refractoriness, which determines the completion of reproductive activity [14]. In some species, e.g., sheep and deer, the thyroid hormones have a seasonal secretion rhythm, with maximum levels preceding the end of the breeding season [15,16]. Seasonal variations in thyroid hormones are known to influence the steroidogenic [17] and spermatogenic function of the testis [18,19], as well as the secretory activity of the epididymis [20] and sexual accessory glands [21]. Thus, the composition of the sperm plasma membrane might be affected by thyroid hormones and influence sperm cryoresistance. Indeed, an association between thyroxine (T4) and sperm cryoresistance has recently been reported in bucks [22].

The mechanistic action of hormones on sperm freezability is unknown. Aquaporins (AQPs) adapt their membrane location to osmotic changes during freezing-thawing—changes that might have a bearing on cryosurvival [23]. AQP expression in sperm cells could thus be involved in the hormone-related seasonal variation of sperm cryoresistance. Certainly, thyroid hormones appear to regulate the expression of AQPs in kidney tissue. For instance, hypothyroid rats show a marked upregulation of renal cortex AQP1 [24], and a significant reduction in the expression of AQP2, AQP3, and AQP4 in the collecting ducts, problems reversible by thyroid hormone replacement [25]. It could be, therefore, that thyroid hormones also regulate AQP expression in sperm cells. 

Males of most wild ruminant species show a short period with high testosterone concentrations in the pre-rutting and rutting season. Ibex and mouflon males were chosen as animal models because they have a similar pattern of testosterone secretion and testicular activity. Therefore, the results obtained could be representative for most of the wild bovid species of the Mediterranean area. The aims of the present study were: (i) to assess the presence of AQPs (i.e., those that allow transmembrane passage of water and neutral solutes, such as glycerol) in the sperm of two wild ruminant species with a marked seasonal reproductive activity: the mouflon and ibex; (ii) to examine the influence of the interaction between photoperiod and T4 on testosterone secretion and AQP expression; and (iii) to determine whether changes in cryoresistance in frozen-thawed sperm are associated with AQP expression. 

## 2. Results

For the ibexes, the infusion of T4 by the osmotic pumps, the melatonin treatment, and their combination (all over January–February), increased (*p* < 0.001) the plasma concentration of T4 (Figure 1A) within the physiological range. The plasma concentration of testosterone also increased (*p* < 0.01) during this period in both the MEL group and MEL+T4 group (Figure 1B). The T4+MEL treatment improved the percentage of fresh sperm with an intact acrosome. The intact acrosome CR was lower (*p* < 0.05) in the MEL group than the control and T4 group, while the CR for sperm viability (*p* < 0.05) was lower in the MEL group than in the T4 group (Table 1).

Western blotting (WB) identified the presence of AQP3 as a single band of about 32–35 KDa. AQP7 was detected as one band of 31 KDa, and AQP10 as two bands of approximately 32 KDa and 38 KDa (Figure 2). 

The ICC assay showed AQP3 as located in different regions, including the acrosome (mean ± sem, 23.0 ± 0.2%; range: 14–26% of total sperm), the post-acrosomal region (8.0 ± 3.2%; range: 1–12%), the mid piece (24.3 ± 0.4%; range: 19–27%), principal piece (24.8 ± 0.1%; 23–28%), and the end piece (22.6 ± 0.1%, 19–27%) (Figure 3). AQP7 was only located in the acrosome (75.0 ± 2.1%, 40–100% of sperm), and AQP10 only in the tail (64.4 ± 1.9%, 20–91% of sperm) (Figure 3). 

For the post-acrosome region, the percentage of sperm showing AQP3 expression decreased (*p* < 0.05), but it increased in the midpiece for the MEL and MEL+T4 groups (Figure 4). The greatest (*p* < 0.01) expression of AQP7 in the acrosome was seen in the ibexes that received melatonin implants (Figure 5). The percentage of sperm showing AQP10 expression in the mid piece was greater (*p* < 0.01) in the MEL than in the control group (Figure 6), while the percentage of sperm with AQP10 expression in the principal piece and end piece was greater (*p* < 0.05) in the MEL+T4 group than in the control group (Figure 6). AQP9 was not detected in any group (or species) either by WB or ICC in our defined experimental conditions.

In the mouflons, the PTU treatment reduced (*p* < 0.001) the T4 plasma concentrations within the physiological range (Figure 7A). No treatment influenced the testosterone secretion over the experimental period (Figure 7B). For the fresh mouflon sperm, the PTU treatment reduced (*p* < 0.05) the percentages of motile and viable sperm, and the percentage of sperm cells with an intact acrosome. Neither PTU treatment, LD treatment, or their combination affected the CR of any sperm variable (Table 2). WB identified AQP3 as a band of 32–36 KDa, AQP7 as one band of 31 KDa, and AQP10 as two bands of approximately 32 KD and 38 KDa, as described above for the ibexes (Figure 2). 

Similarly to that seen for the ibexes, ICC detected AQP3 in different regions of the mouflon sperm, including the acrosome (mean ± sem, 22.0 ± 0.2%; range: 17–27% of total sperm), post-acrosomal region (9.8 ± 0.7%; range 1–21%), midpiece (23.2 ± 0.2%; range: 20–25%), principal piece (23.2 ± 0.2%; range: 21–26%) and endpiece (21.9 ± 0.2; 19–25%) (Figure 3); AQP7 was found only in the acrosome (69.5 ±2.8%; range: 38–98% of total sperm), whereas AQP10 was detected only in the tail (48.0 ± 3.1%; range: 16–82%) (Figure 3). For the post-acrosome region, the percentage of sperm showing AQP3 increased (*p* < 0.05), but it decreased (*p* < 0.05) in the tail (midpiece, principal piece, and end piece) of the LD+PTU group sperm (Figure 8). 

Compared with the PTU group, the percentage of sperm with AQP7 in the acrosome decreased in the group under artificial LD (*p* = 0.05), as it did in the LD+PTU group (*p* < 0.01) (Figure 9). The percentage of sperm with AQP10 in the principal piece and the endpiece of the tail was lower in the PTU+LD group than in the control (*p* = 0.05) and LD (*p* < 0.05) groups (Figure 10).

## 3. Discussion

The results revealed that the photoperiod signal affects the expression of AQP3, AQP7, and AQP10 in the frozen-thawed sperm of wild ruminants. The increase in testosterone concentration due to the melatonin treatment (in the ibex) was associated with reduced sperm freezability. 

In bull spermatozoa, AQP3 and AQP7 have been detected in the midpiece and in the midpiece and post-acrosomal region, respectively [26]. In the present study, however, AQP3 was detected in all sperm regions of both species, whereas AQP7 was exclusively detected in the acrosome. AQP10, which was detected in ibex and mouflon sperm tails, has not been detected in any bovid species. Despite that AQP7 was found only in the acrosome, whereas AQP10 was detected only in the tail, the percentage of sperm with immunolabeling of these AQPs covered a wide range of about 38–100% of sperm for AQP7 and 16–91% of sperm for AQP10. AQP9 was not detected in the present work at any time in either species. Whereas the expression of AQP9 in the efferent ducts and epididymis is well established, and its role in fluid absorption from the tubule lumen accepted [27], no immunohistochemical evidence of its presence in luminal spermatozoa has ever been reported. Although AQP9 was not detected either by WB or ICC in our defined experimental conditions, other antibody concentrations and experimental conditions should be tested. WB detected AQP10 in two bands, suggesting the existence of two isoforms, as previously reported for the human small intestine [28].

The photoperiod signal—induced either via an artificial photoperiod or through melatonin treatment—affected the expression of AQP3, AQP7, and AQP10 in the frozen-thawed sperm of both present species. Melatonin treatment increased the percentage of sperm with AQP7 in the acrosome, and of AQP3 and AQP10 in the midpiece. It was also associated with a higher plasma testosterone concentration, which could have improved reproductive function [29]. The influence of photoperiod/melatonin on AQP expression might be indirectly exerted via changes in the testosterone concentration, which in turn might be associated with reduced sperm cryoresistance. Indeed, it is known that high testosterone levels negatively affect sperm cryoresistance in both the examined species [30]. The present findings suggested that the negative influence of testosterone on sperm freezability may be mediated via AQP expression. The greater expression of AQP3 and AQP10 in the sperm midpiece, and of AQP7 in the acrosome in ibexes treated with melatonin, was associated with a lower cryoresistance ratio for sperm viability and acrosome integrity. Previous studies have shown higher expressions of AQP3 and AQP7 to improve the cryotolerance of boar spermatozoa [31], higher AQP7 expression [26] to improve that of bull spermatozoa, and higher AQP3 to improve the freezability of stallion sperm [32]. Given that the osmolality of the cryopreservation media is high, and that osmotic changes have a detrimental impact on the function and survival of frozen-thawed spermatozoa, it has been suggested that the presence of high AQP3 concentrations could improve the osmoadaptation ability of boar spermatozoa [31]. In contrast, the present data showed that a greater expression of AQP3 in the midpiece, and of AQP7 in the acrosome (possibly induced by testosterone), to be associated with a reduction in sperm freezability for both the examined species. It may be that, despite the AQPs allowing for better sperm osmoadaptation, very high expressions over the short rutting season might lead to abnormal solute permeation [33] with a negative effect on freezability. Moreover, a greater expression of AQP3 and AQP7 might favor a rapid water flux and thus major osmotic stress. Further studies are needed to better understand this discrepancy.

In the present study, AQP expression was only studied in frozen-thawed sperm. However, no differences were reported between fresh and frozen-thawed bull sperm in terms of AQP3 and AQP7 distribution [34]. Similar distributions in fresh samples of ibex and mouflon sperm might therefore be expected. The manipulation of T4 levels, increased by direct infusion or reduced by PTU treatment, only influenced AQP3, AQP7, and AQP10 expression when combined with manipulation of the photoperiod. The data suggest that short days, simulated in the present study via melatonin implants, favored AQP7 and AQP10 expression, whereas their concentrations were reduced under inhibitory long day conditions. Similarly, sperms with AQP3 in the midpiece increased in the ibexes treated with melatonin, and decreased in mouflons in the LD+PTU group. These changes may be related to the capacitation of sperm cells and their bioenergetic requirements during the rutting season. Since AQP7 was only located in the acrosome, a specific role in exocytosis cannot be ruled out. The location of AQP3 and AQP10 in the midpiece, which is packed with mitochondria performing oxidative phosphorylation, and in the principal piece where glycolysis occurs [35] suggest they may be involved in the passage of solutes (e.g., glycerin, lactate, and others) associated with sperm bioenergetic metabolism and motility. Many studies have shown a close association to exist between sperm internal ATP levels and motility, the flagellum beat frequency, and swimming velocity [36]. Short days signal the onset of the rutting season and might favor a greater substrate oxidation capacity by favoring AQP10 and AQP3 expression in the midpiece; higher sperm respiratory or glycolytic rates would synthesize ATP much more quickly. Some AQPs allow the transmembrane passage of both water and neutral solutes, such as glycerol and lactate, the concentrations of which vary with metabolic states [37]. Indeed, a physiological role for AQPs in supporting germ cell metabolism and in preventing energetic imbalances has been suggested [38]. 

Melatonin, along with T4, increased the percentage of ibex sperm with an intact acrosome. The reduction in thyroid hormones after PTU treatment, however, negatively affected all fresh sperm variables in the mouflons, revealing the importance of thyroxine on sperm functionality [39]. No effects on freezability were seen after modifying the secretion of thyroid hormones in both ibexes and mouflons, and the present results did not support them having a role in regulating the end of the rutting season. The melatonin implants, however, prolonged the period of testicular activity (in terms of high testosterone secretion), confirming their influence on the breeding activity of ruminants [40]. 

In conclusion, the photoperiod plays a key role in seasonal AQP expression and reproductive function in wild ruminants. The negative influence of testosterone on sperm cryoresistance might be mediated, at least in part, by an increase in AQP3, AQP7, and AQP10 expression in the acrosome and midpiece during the rutting season. 

## 4. Materials and Methods

### 4.1. Animals

Ejaculates were collected from Iberian ibexes and mouflons at the Animal Reproduction Department of the Spanish National Institute for Agricultural and Food Research and Technology (INIA-CSIC, Madrid, Spain; latitude 40° N). Animal handling procedures were approved by the INIA Ethics Committee (Reference: PROEX 154/17) following European Union Directive 2010/63/UE. 

### 4.2. Experimental Procedure

Marked monthly changes in testosterone secretion are observed over the year in ibexes and mouflons, with baseline levels from January to August, a rise in September, peak concentrations in October and November, and a strong reduction in January to return to basal levels. Taking into account that greatest sperm cryoresistance has been found at the end of the rutting season when testosterone levels fall, in the present work, thyroxine and photoperiod signals were modified during the periods in which testosterone secretion is high, and when basal levels are attained. 

Experiment 1: Influence of thyroxine infusion and melatonin implants on reproductive activity and AQP expression in ibexes. This experiment lasted from December to April. Blood samples for testosterone and thyroxine analysis were taken weekly; sperm samples were collected 40–60 days after the end of the treatments (two semen samples per animal)—the estimated time needed for spermatogenesis to complete. Ibexes were randomly distributed into four groups: (1)Thyroxine group (T4 group): composed of four ibexes kept under natural photoperiod conditions (natural variations in day length from 15 h light/day at the summer solstice to 9 h/day at the winter solstice) that received a continuous infusion of T4 from 1 January to 25 February (i.e., coinciding with the period of natural reduction in the blood plasma testosterone concentration [41]), using Model 2ML2 2 mL Alcet^®®^ osmotic pumps (Durect Corporation, Cupertino, CA, USA). These were implanted (under anesthesia) under the skin in the lateral shoulder area and contained 2.8 mg of T4 (L-Thyroxine T1775, Lot BCBV1017) (Sigma-Aldrich, St. Louis, MO, USA), 0.1 g BSA, 10.6 mg Na_2_CO_3_, and 40 µL NaOH (1N) dissolved in 2.0 mL normal saline solution (0.9%). They delivered 164 µg/day of T4 for 14 days (5.0 µL/h, 14 days), and therefore had to be replaced three times to cover the total infusion period of 56 days. This protocol was previously tested at our laboratory as a means of increasing plasma T4 concentrations within the physiological range; no pathological hyperthyroidism was induced.(2)Melatonin group (MEL group): composed of four ibexes kept under natural photoperiod conditions but which received two subcutaneous (s.c.) melatonin implants (Melovine^®®^) (Ceva Salud Animal, Barcelona, Spain), each of 18 mg, at the base of an ear on 23 December (the winter solstice). This provided for the continuous release of melatonin at a rate maintaining high daytime levels for about 70 days [42], thus establishing a continuous short day-signal from the winter solstice onward, i.e., the photoperiod that stimulates reproductive activity in small ruminants.(3)Thyroxine + melatonin group (T4+MEL group): composed of four ibexes kept under natural photoperiod conditions but receiving continuous infusion of T4 from 1st January to 25 February via the same pumps as described above, and which received two s.c. melatonin implants (as above) at the base of an ear on 23 December.(4)Control group: composed of four ibexes kept under natural photoperiod.

Experiment 2: Influence of propylthiouracil administration and a long day artificial photoperiod on reproductive activity and AQP expression in mouflons. This experiment lasted from November to March. Blood samples for testosterone and thyroxine analysis were taken weekly; sperm samples were collected 40–60 days after the end of the treatments (two semen samples per animal). Mouflons were randomly distributed into four groups: (1)Propylthiouracil-treated group (PTU group): composed of five mouflons kept under natural photoperiod conditions and administered 35 mg/kg propylthiouracil orally in 10 mL of propylene glycol from 1 November to 31 December. This protocol was previously checked at our laboratory as a means of reducing the plasma concentration of thyroxine.(2)Long day photoperiod group (LD group): composed of four mouflons kept in an open stable exposed to long days of 15 h light:9 h dark (15L:9D, equivalent to the summer solstice photoperiod) from 1 November to 31 December. This photoperiod was regulated using an electric clock that operated fluorescent tubes providing an artificial light intensity of approximately 350 lux at floor level. Artificial long days inhibit reproductive activity in wild ruminants [40].(3)Long day photoperiod group (LD+PTU group): composed of five mouflons kept in an open stable exposed to long days (15L:9D) from 1 November to 31 December, and administered 35 mg/kg PTU orally in 10 mL of propylene glycol from 1 November to 31 December.(4)Control group: composed of five mouflons kept under natural photoperiod conditions and administered 10 mL of propylene glycol orally from 1 November to 31 December.

### 4.3. Collection of Samples and Measurements

Blood samples were collected from the jugular vein in heparinized tubes between 10:00 h and 11.00 h. The collected blood was centrifuged at 1500× *g* for 15 min, the plasma separated, and stored at −20 °C until required for testosterone analysis. 

Ejaculates were collected by the transrectal ultrasound-guided massage of the accessory sex glands (TUMASG) [43]. Animals were anesthetized by 0.5 mg/kg intravenous ketamine hydrochloride (Imalgene-1000) (Rhône Mérieux, Lyon, France), 50 µg/kg detomidine (Domosedan) (Pfizer Inc., Amboise Cedex, France), and 0.5 mg/kg tiletamine-zolazepan (Zoletil-100) (Virbac España SA, Barcelona, Spain) Anesthesia was maintained via isoflurane (Isobavet) (Intervet Schering-Plough Animal Health) and later reversed using 0.7 mg/kg yohimbine hydrochloride (half intravenous and half intramuscular) (Sigma, Zwijndrecht, The Netherlands). The time between semen collections for each animal was 14–15 days. 

### 4.4. Hormone Analyses

Testosterone concentrations were measured by radioimmunoassay in duplicate plasma aliquots (100 μL) as previously described [44]. All samples were analyzed in a single assay. The sensitivity was 0.05 ng/mL. The intra-assay coefficient of variation was 11% (with *n* = 7).

The thyroxine concentration was measured using the Sheep Thyroxine (T4) ELISA Kit (Cusabio Technology LLC, Houston, TX, USA) for the mouflon samples, and the Goat Thyroxine (T4) ELISA Kit, (Cusabio) for the ibex samples. Aliquots of 50 µL of plasma were used according to the instructions of the manufacturer. A Microplate Washer MR-12A (Shenzhen Mindray Bio-Medical Electronics, Shenzhen, China) was used to wash the plates. The optical density of each well was determined by a Microplate Reader MR-96A (Shenzhen Mindray Bio-Medical Electronics) set to a wavelength of 450 nm. The intra and inter coefficients of variation were respectively 3% and 2% for the mouflons, and 10% and 11% for the ibexes.

### 4.5. Sperm Quality Analysis

Sperm concentration was assessed by counting the sperm cells in a Neubauer chamber. Sperm membrane and acrosome integrity were assessed by fluorescence using propidium iodide (PI) (Sigma-Aldrich^®®^, St. Louis, MO, USA) combined with fluorescein isothiocyanate-conjugated peanut (Arachis hypogaea) agglutinin (PNA-FITC) (Sigma-Aldrich^®®^, St. Louis, MO, USA). A total of 200 sperm cells were evaluated per sample using a Nikon Eclipse E200 epifluorescence microscope (Nikon Instruments Inc., New York, NY, USA). Sperm membrane integrity was calculated as the sum of all PI-negative cells whereas acrosome integrity was calculated as the sum of all PNA-negative cells. Sperm motility and kinematic variables were examined using a computer-assisted sperm analyzer (CASA) system running Sperm Class Analyzer^®®^ v.4.0. software, (Microptic S.L., Barcelona, Spain) and equipped with an A312fc camera (Basler AG, Ahrensburg, Germany). Samples were diluted in the freezing medium and loaded onto 8 µm × 20 µm well Leja^®^ slides (Leja Products B.V., Nieuw-Vennep, The Netherlands). All materials were maintained at 37 °C and a minimum of 500 sperm tracks and three different fields evaluated per sample with the 10× objective (images acquisition rate 50 frames/s). The following sperm kinetic variables were recorded: total motility (%), progressive motility (%), curvilinear velocity (VCL, µm/s), straight line velocity (VSL, µm/s), and average path velocity (VAP, µm/s).

### 4.6. Sperm Cryopreservation

The freezing extender used for the dilution of the ibex sperm contained 313.7 mM Tris (Merck KGaA, Darmstadt, Germany), 104.7 mM citric acid (Merck KGaA, Darmstadt, Germany), and 30.3 mM glucose (Merck KGaA, Darmstadt, Germany). That used for the mouflon samples contained 210.6 mM TES, 95.8 mM Tris, and 10.1 mM glucose. Both types of extenders contained 6% egg yolk (*v*/*v*) and 5% glycerol (*v*/*v*). Sperm samples were cryopreserved by slow freezing in straws using previously optimized techniques [45]. Briefly, sperm was diluted with the freezing extender to a final concentration of 100 × 10^6^ sperm/mL. Diluted sperm was cooled to 5 °C for 180 min. Straws were exposed to liquid nitrogen vapor for the last 10 min before being immersed and stored in liquid nitrogen. Straws were thawed in a water bath at 37 °C for 30 s for the post-thaw quality evaluation. Sperm freezability was assessed by calculation of the cryoresistance ratio (CR) as CR = post-thaw value/fresh value × 100 for the different measured variables.

### 4.7. AQP Assay 

Spermatozoa were examined for the presence and distribution of AQP3, AQP7, AQP9, and AQP10 by Western blotting (WB) and immunocytochemistry (ICC), employing commercial rabbit polyclonal antibodies (AQP3—ab125219, AQP7—ab32826, AQP9—ab191056, and AQP10—ab182794) (all from Abcam (Netherlands) B.V). Controls for the specificity of antibodies were previously established in our lab (Appendix A). Small intestine and kidney of sheep were used to test the specificity of the antibodies (positive control) in WB. Negative controls where the sample was incubated only with secondary antibody, omitting the primary antibody step, were also included in each immunolabeling assay (Appendix A). For WB, proteins were extracted from 35 million spermatozoa; after three centrifugations (at 5400× *g* for 5 min), the pellet was subjected to crude mechanical disruption and incubated with lysis buffer at 4 °C for 60 min. The lysis buffer was composed of 6% sodium dodecyl sulfate (SDS) (Merck KGaA, Darmstadt, Germany), 125 mM Tris (Merck KGaA, Darmstadt, Germany), 1 mM benzamide (Merck KGaA, Darmstadt, Germany), 1/100 (*v*/*v*) protease inhibitor cocktail (Thermo Scientific, Rockford, IL, USA), and 1 mM phenylmethylsulphonyl fluoride (Merck KGaA, Darmstadt, Germany). The samples were then centrifuged at 5400× *g* for 5 min, the supernatant collected, and Laemmli sample buffer (DTT, SDS, Tris, glycerol, b-mercaptoethanol, and bromophenol blue) added. These protein suspensions were then denatured by heating +94 °C for 4 min, and aliquots of 35 µL loaded on 12% SDS-PAGE gels. Electrophoresis was performed at 150 V for 90 min, followed by the transfer of the proteins to Amersham™ Protran^®®^ 0.45 µm nitrocellulose membranes (Merck KGaA, Darmstadt, Germany) at 300 mA for 90 min, blocking with 5% BSA in PBS-Tween for 60 min, and incubation overnight at +4 °C with a dilution 1/1000 of the primary antibodies. The membranes were then washed three times in PBS-Tween, and incubated with the secondary antibody (mouse anti-rabbit IgG-HRP sc-2357) (Santa Cruz Biotechnology Inc., Dallas, TX, USA) with a dilution 1/15,000 for 120 min, followed by extensive washing in PBS-Tween. The membranes were scanned using WesternSure^®®^ PREMIUM, LI-COR^®®^ chemiluminiscent substrate (Lincoln, NE, USA), employing an Amersham™ ECL Western Blotting ImageQuant™ 500 chemiluminiscent imaging system (Ge Healthcare). 

For ICC, spermatozoa were fixed in 4% paraformaldehyde, centrifuged (1200× *g*, 6 min), and the pellet resuspended in PBS to prepare smears on slides. The smears were allowed to dry, washed with PBS-Tween, and blocked with 5% BSA (Sigma-Aldrich, Sweden) in PBS for 60 min. After washing, the slides were incubated with the primary antibodies against AQPs overnight at 4°C; primary antibodies were diluted 1/100 in PBS containing 0.1% Tween 20 and 1% BSA. The smears were then washed before incubation with the secondary antibody (polyclonal goat anti-rabbit Alexa Fluor 488) (Molecular Probes, Invitrogen, Carlsbad, CA, USA), diluted 1/500 in PBS containing 0.1% Tween 20 and 1% BSA, in darkness for 180 min [46]. The sperm membrane location of the AQPs was checked by confocal microscopy using a Zeiss LSM800 inverted confocal laser scanning microscope at ×630 magnification, running NIS software. In addition, the percentage of sperms showing AQPs in different regions of cell was evaluated using a Nikon Eclipse E200 epifluorescence light microscope (Nikon Instruments Inc., New York, NY, USA), examining 200 cells.

### 4.8. Statistical Analysis 

Data distributions were examined using the Shapiro–Wilk test. The homogeneity of variance was assessed using the Levene test. The influence of the treatments on the sperm variables and on the cryoresistance ratio were analyzed by ANOVA; mean differences between groups were assayed by Fisher’s LSD post-hoc test. To test the effect of the T4 infusion and PTU treatments on plasma T4 concentrations, means were compared by the Student *t*-test. Plasma testosterone concentrations and AQP expression in the different experimental groups were compared using the same *t* test. All statistical analyses were performed using STATISTICA software for Windows v.12.0 (StatSoft, Inc., Palo Alto, CA, USA).

## Figures and Tables

**Figure 1 ijms-23-02903-f001:**
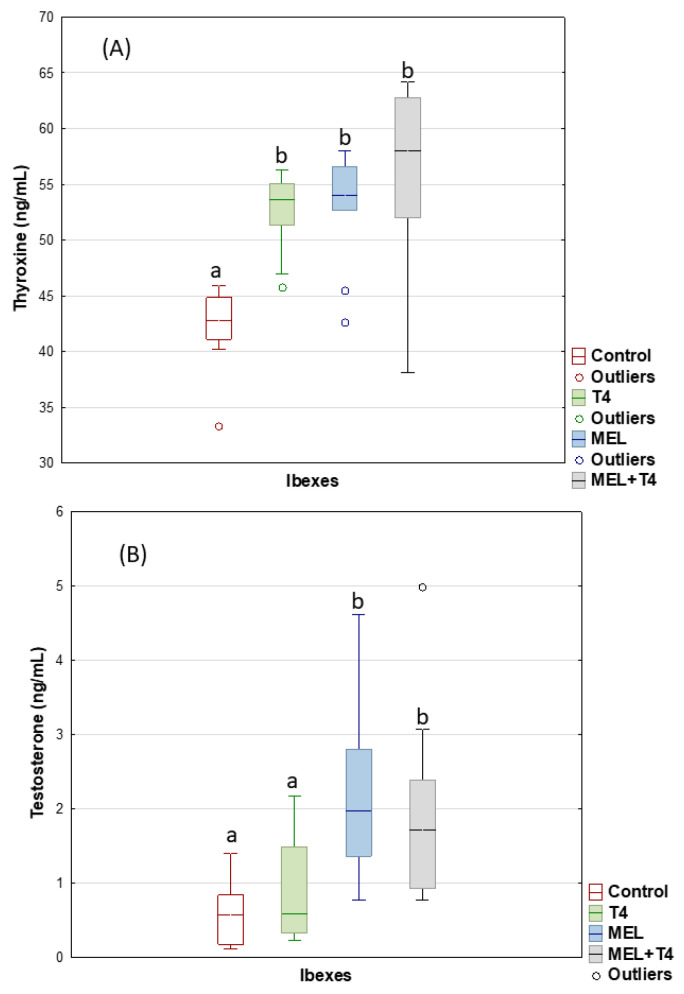
Plasma thyroxine (**A**) and testosterone (**B**) concentrations in ibexes: Controls (red); treated with thyroxine (T4, green); treated with melatonin implants (MEL, blue); and treated with melatonin implants plus thyroxine (MEL+T4, gray). Box plots show the median (horizontal line) and spread from the 1st to the 3rd quartiles; the whiskers extend from the smallest to the largest value. Outliers are shown as circles. Different letters (a,b) between boxplots indicate significant differences (*p* < 0.001 for (**A**) and *p* < 0.01 for (**B**)).

**Figure 2 ijms-23-02903-f002:**
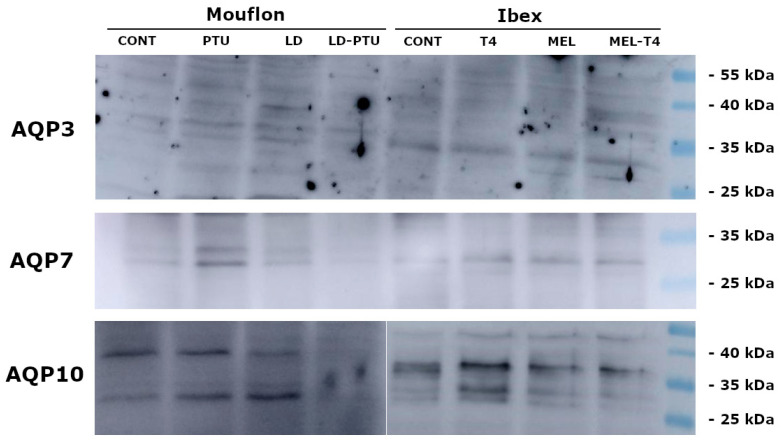
Western blot images showing expression patterns of AQP3, AQP7, and AQP10 in ibex sperm (controls treated with thyroxine (T4), treated with melatonin implants (MEL), treated with melatonin implants plus thyroxine (MEL+T4)), and mouflon sperm (controls administered propylthiouracil (PTU), exposed to long day photoperiod (LD), exposed to long day photoperiod plus PTU (LD+PTU)).

**Figure 3 ijms-23-02903-f003:**
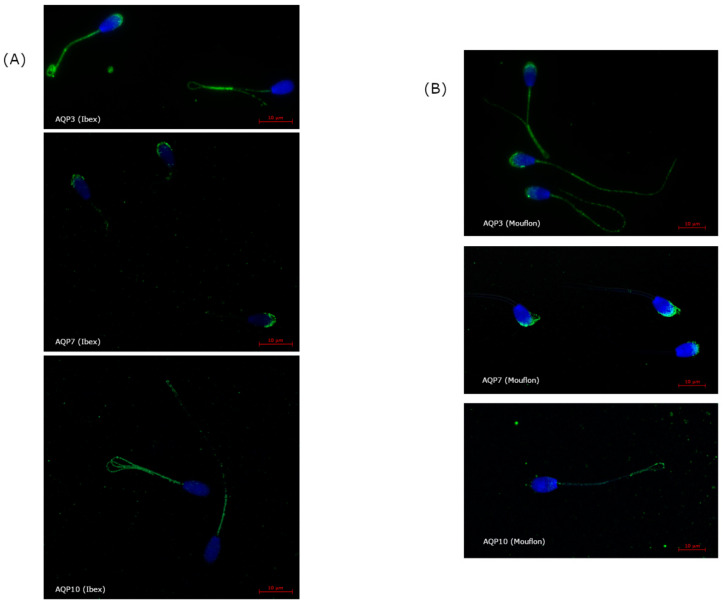
Immunolabeling of AQP3 (located in the acrosome, post-acrosomal region, midpiece, principal piece, and end piece), AQP7 (located in the acrosome), and AQP10 (located in the principal piece and the endpiece) in ibex sperm (**A**), and in mouflon sperm (**B**).

**Figure 4 ijms-23-02903-f004:**
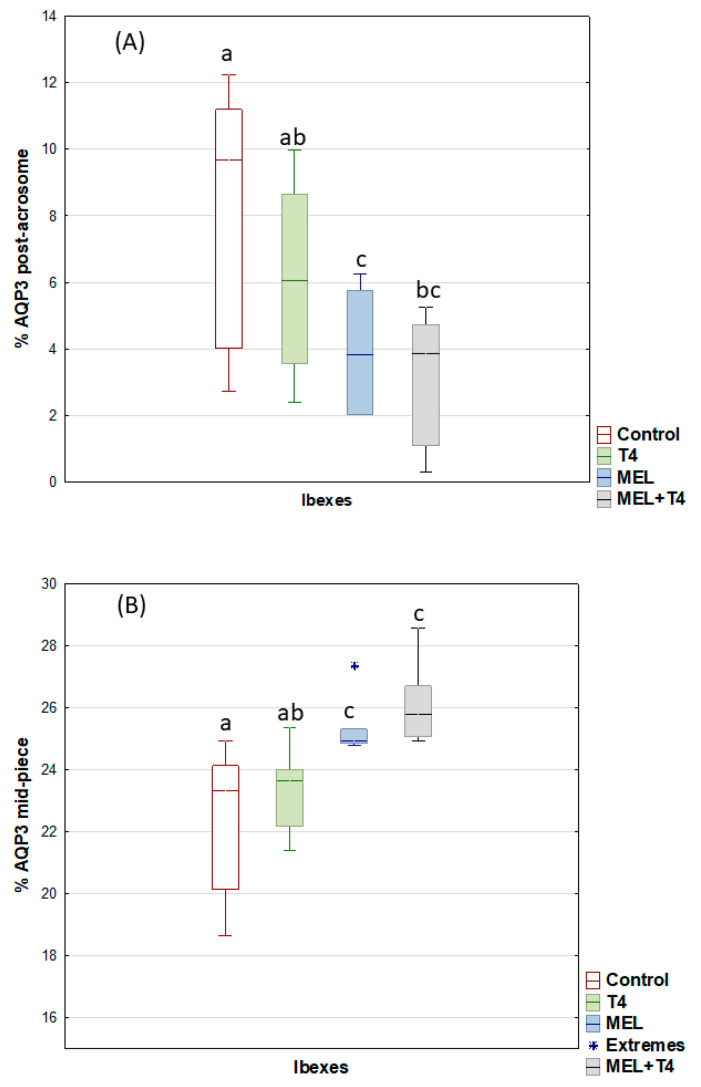
AQP3 expression as determined by immunocytochemistry labelling (ICC) in the post-acrosome (**A**) and midpiece (**B**) of ibex sperm (controls (red); treated with thyroxine (T4, green); treated with melatonin implants (MEL, blue); and treated with melatonin implants plus thyroxine (MEL+T4, gray)). The boxes spread form the 1st to the 3rd quartiles. Box plots show the median (horizontal line) and spread from the 1st to the 3rd quartiles; the whiskers extend from the smallest to the largest value. Extremes are shown as asterisks. Different letters (a,b,c) between boxplots indicate significant differences (*p* < 0.05).

**Figure 5 ijms-23-02903-f005:**
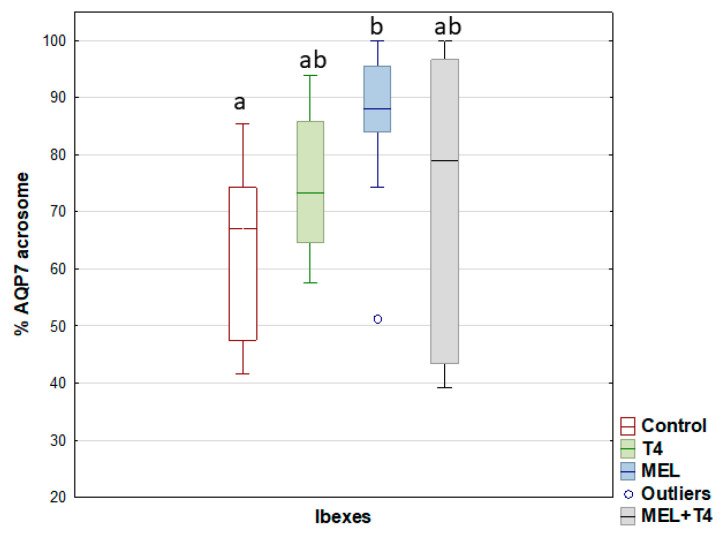
AQP7 expression by immunocytochemistry labelling (ICC) in acrosome of ibex sperm (controls (red); treated with thyroxine (T4, green); treated with melatonin implants (MEL, blue); and treated with melatonin implants plus thyroxine (MEL+T4, gray)). The boxes spread form the 1st to the 3rd quartiles. Box plots show the median (horizontal line) and spread from the 1st to the 3rd quartiles; the whiskers extend from the smallest to the largest value. Outliers are shown as circles. Different letters (a,b) between boxplots indicate significant differences (*p* < 0.01).

**Figure 6 ijms-23-02903-f006:**
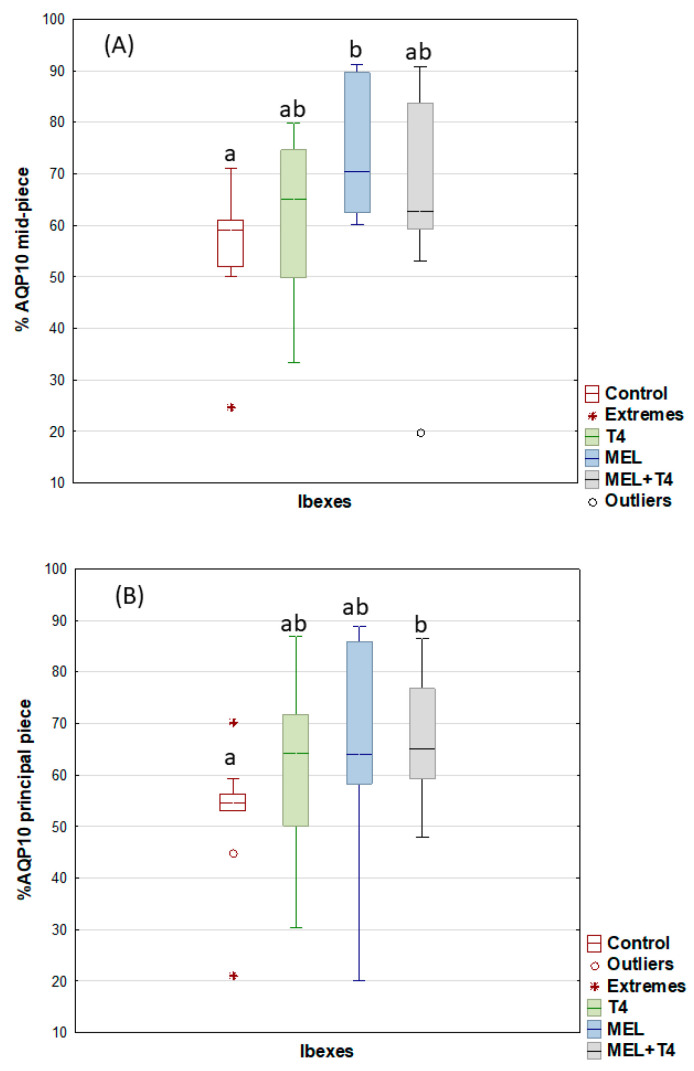
AQP10 expression by immunocytochemistry labelling (ICC) in midpiece (**A**) and principal piece of ibex sperm (controls (red); treated with thyroxine (T4, green); treated with melatonin implants (MEL, blue); and treated with melatonin implants plus thyroxine (MEL+T4, gray)). The boxes spread form the 1st to the 3rd quartiles. Box plots show the median (horizontal line) and spread from the 1st to the 3rd quartiles; the whiskers extend from the smallest to the largest value. Outliers are shown as circles and extremes as asterisks. Different letters (a,b) between boxplots indicate significant differences (*p* < 0.01 for (**A**), and *p* < 0.05 for (**B**)).

**Figure 7 ijms-23-02903-f007:**
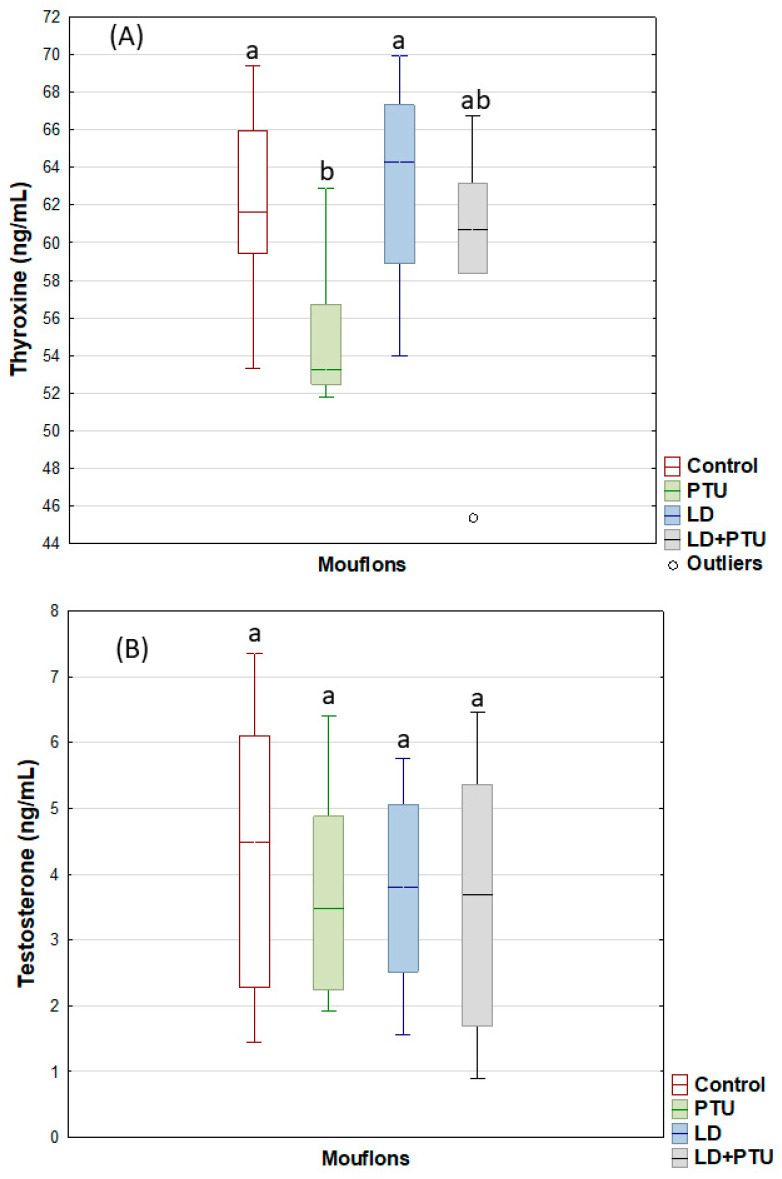
Plasma thyroxine (**A**) and testosterone (**B**) concentrations in mouflons: controls (red), administered propylthiouracil (PTU, green), exposed to long day photoperiod (LD, blue), exposed to long day photoperiod plus administration of PTU (LD+PTU, gray). The boxes spread form the 1st to the 3rd quartiles. Box plots show the median (horizontal line) and spread from the 1st to the 3rd quartiles; the whiskers extend from the smallest to the largest value. Outliers are shown as circles. Different letters (a,b) between boxplots indicate significant differences (*p* < 0.001).

**Figure 8 ijms-23-02903-f008:**
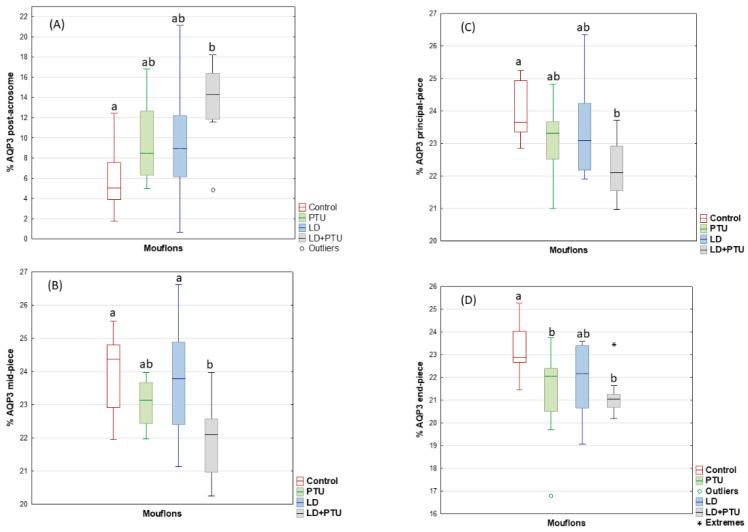
AQP 3 expression by immunocytochemistry labelling (ICC) in post-acrosome (**A**), midpiece (**B**), principal piece (**C**) and end-piece (**D**) of mouflon sperm (controls (red); administered propylthiouracil (PTU, green), exposed to long day photoperiod (LD, blue), exposed to long day photoperiod plus PTU (LD+PTU, gray)). The boxes spread form the 1st to the 3rd quartiles. Box plots show the median (horizontal line) and spread from the 1st to the 3rd quartiles; the whiskers extend from the smallest to the largest value. Outliers are shown as circles and extremes as asterisks. Different letters (a,b) between boxplots indicate significant differences (*p* < 0.05).

**Figure 9 ijms-23-02903-f009:**
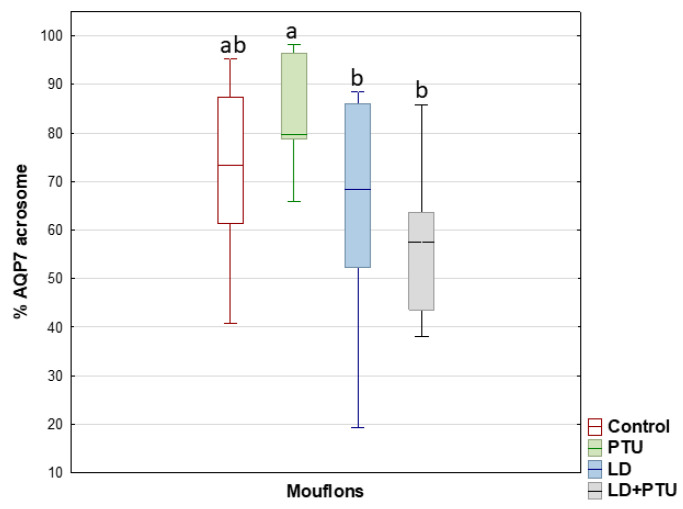
AQP7 expression by immunocytochemistry labelling (ICC) in acrosome of mouflon sperm (controls (red); administered propylthiouracil (PTU, green), exposed to long day photoperiod (LD, blue), exposed to long day photoperiod plus PTU (LD+PTU, gray)). The boxes spread form the 1st to the 3rd quartiles. Box plots show the median (horizontal line) and spread from the 1st to the 3rd quartiles; the whiskers extend from the smallest to the largest value. Different letters (a,b) between boxplots indicate significant differences (PTU vs. LD, *p* = 0.05; PTU vs. LD+PTU, *p* = 0.01).

**Figure 10 ijms-23-02903-f010:**
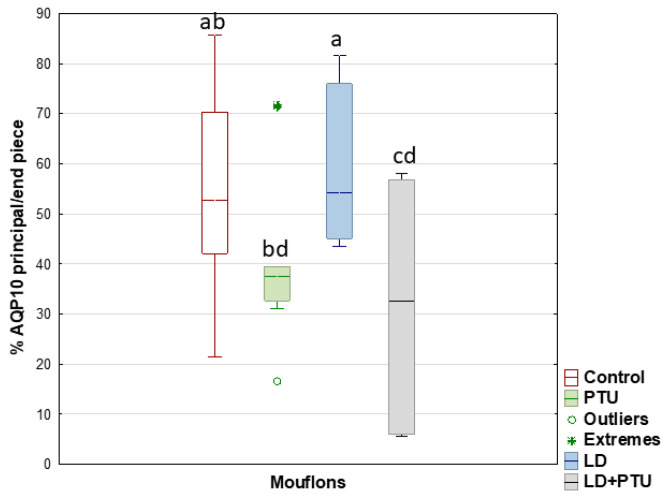
AQP10 expression by immunocytochemistry labelling (ICC) in principal and endpiece of tail of mouflon sperm (controls (red); administered propylthiouracil (PTU, green), exposed to long day photoperiod (LD, blue), exposed to long day photoperiod plus PTU (LD+PTU, gray)). The boxes spread form the 1st to the 3rd quartiles. Box plots show the median (horizontal line) and spread from the 1st to the 3rd quartiles; the whiskers extend from the smallest to the largest value. Outliers are shown as circles and extremes as asterisks. Different letters (a,b,c,d) between boxplots indicate significant differences (LD+PTU vs. Control, *p* = 0.05; LD+PTU vs. LD, *p* < 0.05).

**Table 1 ijms-23-02903-t001:** Ibex fresh sperm variables and cryoresistance ratios (CR) for the control, T4 (treated with thyroxine), MEL (treated with melatonin implant), and T4+MEL (treated with thyroxine plus melatonin implant) groups. Different letters indicate significant differences (*p* < 0.05).

Fresh Sperm Variables	Control	T4	MEL	T4+MEL
Motile sperm (%)	41.7 ± 8.7	51.9 ± 7.9	59.3 ± 14.1	64.4 ± 7.3
Intact acrosome (%)	69.0 ± 15.7 a	79.0 ± 11.6 ab	88.1 ± 5.1 ab	96.78 ± 2.1 b
Viable sperm (%)	42.8 ± 13.2	55.9 ± 10.5	56.1 ± 6.7	56.8 ± 7.2
**Cryoresistance ratio**				
CR-Motile sperm	53.3 ± 5.6	61.0 ± 12.4	38.9 ± 7.3	44.3 ± 5.7
CR-Intact acrosome	89.1 ± 15.3 a	75.5 ± 4.7 a	49.5 ± 11.0 b	64.8 ± 3.1 ab
CR-Viable sperm	51.7 ± 7.1 ab	57.1 ± 13.0 a	28.5 ± 6.4 b	48.7 ± 9.0 ab

**Table 2 ijms-23-02903-t002:** Mouflon fresh sperm variables and cryoresistance ratio (CR) for the control, PTU (treated with propylthiouracil), LD (artificial long day photoperiod) and LD+PTU (artificial long day photoperiod plus PTU) groups. Different letters indicate significant differences (*p* < 0.05).

Fresh Sperm Variables	Control	PTU	LD	LD+PTU
Motile sperm (%)	66.6 ± 5.7 a	51.5 ± 5.8 b	73.1 ± 4.3 a	77.1 ± 3.4 a
Intact acrosome (%)	90.4 ± 2.8 a	79.6 ± 5.2 b	92.7 ± 3.1 a	97.3 ± 0.8 a
Viable sperm (%)	68.7 ± 3.3 a	47.1 ± 8.0 b	67.1 ± 4.0 a	77.4 ± 4.0 a
**Cryoresistance ratio**				
CR-Motile sperm	45.9 ± 9.8	45.6 ± 6.7	28.3 ± 8.0	37.9 ± 10.0
CR-Intact acrosome	67.7 ± 9.1	70.8 ± 3.9	59.3 ± 6.8	67.4 ± 6.8
CR-Viable sperm	32.7 ± 8.1	37.6 ± 7.8	22.0 ± 6.1	32.1 ± 13.0

## Data Availability

The data that support the findings of this study are available on request from the corresponding author.

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
