# Peer review of "Expression of Aquaglyceroporins in Spermatozoa from Wild Ruminants Is Influenced by Photoperiod and Thyroxine Concentrations"

_ijms, 2022, doi:10.3390/ijms23062903_

Round 1
Reviewer 1 Report
In the present manuscript, the authors studied in wild ibexes and mouflons sperm cells: the expression of aquaglyceroporins (AQPs), the effect of the interaction between photoperiod (melatonin) and T4 on testosterone secretion and AQP expression, the changes in cryoresistance related to AQPs expression. Results showed that sperms expressed AQP3, 7 and 10, localized in different specific regions. The photoperiod (melatonin) regulates the seasonal expression of AQP and reproductive function in wild ruminants. Finally, testosterone was found to reduce the sperm cryoresistance probably by an increase in AQP3, AQP7 and AQP10 expression in sperm during the rutting season.
The study is of clear design, the methodology is presented in sufficient detail, and the results are well described and discussed. However, there are major concerns that the authors need to address.
Comments
- In the figures legends, the statistical significances were different from those in the text (i.e. in Figure 1 legend “…Different letters indicate significant differences (p < 0.01).” while in the text was indicated “…increased (p < 0.001) the plasma…”.). All discrepancies need to be corrected.
- Line 95; western blot should be abbreviated here and not at line 397. Blots are of poor quality, especially that of AQP3. Moreover, a semi-quantitative analysis (normalized with housekeeping) of the bands could show up- or downregulation of the AQPs.
- Negative controls of western blot and ICC experiments should be added, at least in the supplemental materials.
- Figure 4 legend; “outliers are shown as circles” but “Extremes” are indicated in the figure as an asterisk. Similarly, in Figure 6 “Outliers” and “extremes” are indicated while only “outliers” were described in the legend. This should be amended or made uniform.
- Table 2 cited at line 147 is not present in the article.
- Lines 171-175. Why did the authors compare the LD and LD+PTU vs PTU and not vs controls? Moreover, was PTU significantly modified?
- Line 181, Outliers are not present in the figure. Please, the sentence in the legend should be deleted.
- Line 189, extremes are not described in the legend.
- Line 389, “100 x 106sperm/mL…” change with 10^6 (superscript).
Author Response
Reply to Reviewer 1
We thank the referee for his/her work and helpful recommendations. We have considered all the points raised and have made the appropriate corrections. We hope you will find these changes have improved the manuscript.
In the present manuscript, the authors studied in wild ibexes and mouflons sperm cells: the expression of aquaglyceroporins (AQPs), the effect of the interaction between photoperiod (melatonin) and T4 on testosterone secretion and AQP expression, the changes in cryoresistance related to AQPs expression. Results showed that sperms expressed AQP3, 7 and 10, localized in different specific regions. The photoperiod (melatonin) regulates the seasonal expression of AQP and reproductive function in wild ruminants. Finally, testosterone was found to reduce the sperm cryoresistance probably by an increase in AQP3, AQP7 and AQP10 expression in sperm during the rutting season.
The study is of clear design, the methodology is presented in sufficient detail, and the results are well described and discussed. However, there are major concerns that the authors need to address.
Comments
- In the figures legends, the statistical significances were different from those in the text (i.e. in Figure 1 legend “…Different letters indicate significant differences (p < 0.01).” while in the text was indicated “…increased (p < 0.001) the plasma…”.). All discrepancies need to be corrected.
[Authors' response: Correction made. Statistical significances have been revised through the manuscript].
- Line 95; western blot should be abbreviated here and not at line 397.
[Authors' response: Correction made].
Blots are of poor quality, especially that of AQP3. Moreover, a semi-quantitative analysis (normalized with housekeeping) of the bands could show up- or downregulation of the AQPs.
[Authors' response: Unfortunately, we do not have more samples to replicate WB analyses. In our experimental design, we include a semi-quantitative quantification of the expression by ICC of AQP3 (Figure 4, Figure 8), AQP7 (Figure 5, Figure 9) and AQP10 (Figure 6, Figure 10). In addition, we have confirmed the presence of AQPs by WB identification, by extracting the same amount of sperm (35 million) and loading in each lane of the SDS-PAGE gel, as depicted in Figure 2. As long as our experimental design focuses not only on the expression changes but also the locations, the ICC offers the possibility of analyzing both measurements using the same technique. However, we understand the reviewer's concern and we would definitely take into account your advice of using endogenous controls in future experimental designs, including the overall changes in protein expression].
- Negative controls of western blot and ICC experiments should be added, at least in the supplemental materials.
[Authors' response: Negative controls of WB and ICC have been included as Supplementary Figures 1 and 2, in revised version].
- Figure 4 legend; “outliers are shown as circles” but “Extremes” are indicated in the figure as an asterisk. Similarly, in Figure 6 “Outliers” and “extremes” are indicated while only “outliers” were described in the legend. This should be amended or made uniform.
[Authors' response: Correction made. We have harmonised the Figure legends throughout the manuscript].
- Table 2 cited at line 147 is not present in the article.
[Authors' response: Sorry for the mistake. The table 2 has been included in the article].
- Lines 171-175. Why did the authors compare the LD and LD+PTU vs PTU and not vs controls? Moreover, was PTU significantly modified?
[Authors' response: LD group was compared to LD+PTU and PTU groups, but all groups were also compared to control group. We have only highlighted the significant results in this paragraph. Figure 9 shows that there are no differences between the control group and the remaining experimental groups].
- Line 181, Outliers are not present in the figure. Please, the sentence in the legend should be deleted.
[Authors' response: Correction made].
- Line 189, extremes are not described in the legend.
[Authors' response: Correction made].
- Line 389, “100 x 106sperm/mL…” change with 10^6 (superscript).
[Authors' response: Correction made].
Reviewer 2 Report
In this paper, Santiago-Moreno and colleagues studied the effects of photoperiod and thyroxine on aquaporins expression and localization in ibexes and mouflons frozen-thawed sperm. They found that melatonin increased testosterone concentration in the ibexes, while it reduced the cryoresistance ratio (CR) for sperm viability and the presence of a normal acrosome; moreover, it induced localization of AQP7 in the acrosome and of AQP3 and AQP10 in the midpiece.
In the mouflons, the artificial long day and reduced thyroxine increased the percentage of sperm with AQP10 in the principal piece.
The authors concluded that photoperiod/melatonin may indirectly affect AQPs expression influencing testosterone concentration.
The paper is well conducted; however, some changes should be addressed:
- The authors should specify why different experimental approaches were used for the two species
- A control for the procedure, namely animals infused with saline, should have been added
- The authors assessed that AQP7 was only located in acrosome, while AQP10 in the tail; however, the given percentages cover a very wide range (40-100% and 20-91%, respectively), thus, the “specific” localization is not so evident
- In Figure 3, a scale bar should be added, and the picture representing AQP7 (Mouflon) should be replaced, or the background adjusted
- More details concerning the dilution of the used antibody and the composition of the lysis buffer should be given; moreover, the concentration of the loaded protein samples should be specified
- The authors should better specify why, in control, CR-viable sperm are more that in fresh samples
Author Response
Reply to Reviewer 2
We would like to thank this referee for her/his thorough examination of our paper. We have carefully considered the comments made and we have introduced the suggested corrections.
The paper is well conducted; however, some changes should be addressed:
- The authors should specify why different experimental approaches were used for the two species
[Authors' response: Ibex and mouflon males were chosen as animal models because they have a similar pattern of testosterone secretion and testicular activity. Therefore, the results obtained could be representative for most of the wild bovid species of the Mediterranean area (we have clarified this fact in the Introduction section of the revised version). In addition, because the possibility of work with a large number wild animals is very limited, the inclusion of two different species could increase the scope and number of the experimental designs].
- A control for the procedure, namely animals infused with saline, should have been added
[Authors' response: We considered different options for control groups at the beginning of the experiments. In the Experiment 1, ibexes were implanted with Alcet® osmotic pumps (T4 group), receiving subcutaneous melatonin implants (MEL group) or receiving both implants and osmotic pumps. Certainly, in this experiment at least a group receiving Alcet® osmotic pumps with saline should have been added, as suggested by referee, but limitation in the number of animals and welfare reasons prevented the inclusion of an additional group. Experiment facilities for wild species allows work to be carried out with a very limited number of animals. Regarding the experiment 2, we have completed the description of control group in the revised version: “Control group: composed of five mouflons kept under natural photoperiod conditions and administered 10 ml of propylene glycol orally from November 1st to December 31th” (L304-305)].
- The authors assessed that AQP7 was only located in acrosome, while AQP10 in the tail; however, the given percentages cover a very wide range (40-100% and 20-91%, respectively), thus, the “specific” localization is not so evident
[Authors' response: We want to highlight that, in both species, AQP7 was found only in the acrosome, whereas AQP10 was detected only in the tail, but not in other sperm regions as observed in AQP3. Certainly, the percentages of spermatozoa cover a very wide range, and we have noted this fact in L192-194 of the revised version].
- In Figure 3, a scale bar should be added, and the picture representing AQP7 (Mouflon) should be replaced, or the background adjusted
[Authors' response: Correction made].
- More details concerning the dilution of the used antibody and the composition of the lysis buffer should be given; moreover, the concentration of the loaded protein samples should be specified
[Authors' response: Details about the dilution of the used antibody is provided as suggested by referee in the revised version. Composition of the lysis buffer is given in M&M section of the revised version. We standardize the exactly same amount of sperm cells per sample (35 million), and per animal, so that the total number of cells yield similar concentrations of total protein. Unfortunately, we did not measure the total protein concentration after the buffer lysis extraction but we will take into consideration your suggestion for future analysis].
- The authors should better specify why, in control, CR-viable sperm are more that in fresh samples
[Authors' response: Viable sperm (%) refers to the percentage of spermatozoa showing membrane integrity On the other, hand cryoresistance ratio for viable sperm is a value (no percentage) assessed by calculation of the following formula: CR = post-thaw value/fresh value x 100 for the different measured variables (see L355-356 in revised version)].
Reviewer 3 Report
The authors provide an interesting analysis of expression of AQPs in frozen-thawed sperm in wild ruminants. The study has are some weak points both in the design, methodology and results.
Comments can be found below:
The authors should include the mean of percentages of each distribution pattern for AQPs.
In Figure 3 is necessary to add the scale bars.
AQP9 was not detected in any groups experimental. Have been tested different concentrations and experimental conditions? The authors should performance this experiments to to justify the results.
The graphs of figures 4, 5 and 6 should be illustrated with the same scale.
In Figure 10, add the meaning of letters a,b, c and d (foot figure).
In discussion the authors say “Similar distributions in fresh samples of ibez and mouflon sperm might therefore be expected”. Why haven’t they studied the AQPs in fresh sperm? These experiments are necessary to understand the results in this study.
What controls have been used for Acrosme Reaction?
The controls groups in Experiment 1 and 2 should be explained in more detail.
Author Response
Reply to Reviewer 3
We thank the referee for his/her work and helpful suggestions. We have considered all the points raised and have made the appropriate corrections.
The authors provide an interesting analysis of expression of AQPs in frozen-thawed sperm in wild ruminants. The study has are some weak points both in the design, methodology and results.
Comments can be found below:
The authors should include the mean of percentages of each distribution pattern for AQPs.
[Authors' response: Correction made (see L99-102 and L156-159 of the revised version)].
In Figure 3 is necessary to add the scale bars.
[Authors' response: Correction made].
AQP9 was not detected in any groups experimental. Have been tested different concentrations and experimental conditions? The authors should performance this experiments to justify the results.
[Authors' response: AQP9 was not detected in any group (or species) either by WB or ICC “in our defined experimental conditions” (see L112-113 in revised version). Certainly, we agree that other experimental conditions should be made (see L201-202 in revised version)].
The graphs of figures 4, 5 and 6 should be illustrated with the same scale.
[Authors' response: We have homogenized the dimensions of Figures 4, 5, 6, but the Y-axis scale was adapted to the values found different sperm regions and AQPs for a better clarity].
In Figure 10, add the meaning of letters a,b, c and d (foot figure).
[Authors' response: Correction made. Different letters indicate significant differences (cd vs ab, p = 0.05; cd vs a, p < 0.05). (See foot Figure 10 in revised version)].
In discussion the authors say “Similar distributions in fresh samples of ibex and mouflon sperm might therefore be expected”. Why haven’t they studied the AQPs in fresh sperm? These experiments are necessary to understand the results in this study.
[Authors' response: We agree with referee. Because the limitation on sperm collection in wild ruminants, the implementation of AQPs analysis by WB and ICC was made with frozen-thawed samples. We understand the reviewer's concern and we would definitely take into account your advice of using fresh samples in future experimental designs].
What controls have been used for Acrosme Reaction?
[Authors' response: Acrosome integrity were assessed by fluorescence using fluorescein isothiocyanate-conjugated peanut (Arachis hypogaea) agglutinin (PNA-FITC) (Sigma-Aldrich®, St. Louis, USA), counting a total of 200 sperm cells in each experimental group: Control, T4, MEL, T4+MEL for ibexes, and Control, PTU, LD and LD+PTU for mouflons. Acrosome integrity was analyzed in fresh and frozen-thawed samples, which allowed the calculation of the cryoresistance ratio].
The controls groups in Experiment 1 and 2 should be explained in more detail.
[Authors' response: In the Experiment 1, at least a group receiving Alcet® osmotic pumps with saline should have been added, but limitation in the number of animals and welfare reasons prevented the inclusion of an additional group. Regarding the experiment 2, we have completed the description of control group in the revised version: “Control group: composed of five mouflons kept under natural photoperiod conditions and administered 10 ml of propylene glycol orally from November 1st to December 31th”].
Round 2
Reviewer 1 Report
The manuscript was revised according to almost all of my comments and suggestions.
Reviewer 2 Report
The paper can be accepted in this form.
Reviewer 3 Report
The authors have responded appropriately to the comments and the manuscript has been improved.